# Top-Down Regularization of Deep Belief Networks

**Hanlin Goh**[*]**, Nicolas Thome, Matthieu Cord**
Laboratoire d'Informatique de Paris 6
UPMC – Sorbonne Universités, Paris, France
`{Firstname.Lastname}@lip6.fr`

**Joo-Hwee Lim**[†]
Institute for Infocomm Research
A*STAR, Singapore
`joohwee@i2r.a-star.edu.sg`

## Abstract

Designing a principled and effective algorithm for learning deep architectures is a challenging problem. The current approach involves two training phases: a fully unsupervised learning followed by a strongly discriminative optimization. We suggest a deep learning strategy that bridges the gap between the two phases, resulting in a three-phase learning procedure. We propose to implement the scheme using a method to regularize deep belief networks with top-down information. The network is constructed from building blocks of restricted Boltzmann machines learned by combining bottom-up and top-down sampled signals. A global optimization procedure that merges samples from a forward bottom-up pass and a top-down pass is used. Experiments on the MNIST dataset show improvements over the existing algorithms for deep belief networks. Object recognition results on the Caltech-101 dataset also yield competitive results.

## 1   Introduction

Deep architectures have strong representational power due to their hierarchical structures. They are capable of encoding highly varying functions and capture complex relationships and high-level abstractions among high-dimensional data [1]. Traditionally, the multilayer perceptron is used to optimize such hierarchical models based on a discriminative criterion that models $P(\mathbf{y}|\mathbf{x})$ using a error backpropagating gradient descent [2, 3]. However, when the architecture is deep, it is challenging to train the entire network through supervised learning due to the large number of parameters, the non-convex optimization problem and the dilution of the error signal through the layers. This optimization may even lead to worse performances as compared to shallower networks [4].

Recent developments in unsupervised feature learning and deep learning algorithms have made it possible to learn deep feature hierarchies. Deep learning, in its current form, typically involves two consecutive learning phases. The first phase greedily learns unsupervised modules layer-by-layer from the bottom-up [1, 5]. Some common criteria for unsupervised learning include the maximum likelihood that models $P(\mathbf{x})$ [1] and the input reconstruction error of vector $\mathbf{x}$ [5–7]. This is subsequently followed by a supervised phase that fine-tunes the network using a supervised, usually discriminative algorithm, such as supervised error backpropagation. The unsupervised learning phase initializes the parameters without taking into account the ultimate task of interest, such as classification. The second phase assumes the entire burden of modifying the model to fit the task.

In this work, we propose a gradual transition from the fully-unsupervised learning to the highly-discriminative optimization. This is done by adding an intermediate training phase between the two existing deep learning phases, which enhances the unsupervised representation by incorporating top-down information. To realize this notion, we introduce a new global (non-greedy) optimization

---

[*]Hanlin Goh is also with the Institute for Infocomm Research, A*STAR, Singapore and the Image and Pervasive Access Lab, CNRS UMI 2955, Singapore – France.

[†]Joo-Hwee Lim is also with the Image and Pervasive Access Lab, CNRS UMI 2955, Singapore – France.

that regularizes the deep belief network (DBN) from the top-down. We retain the same gradient descent procedure of updating the parameters of the DBN as the unsupervised learning phase. The new regularization method and deep learning strategy are applied to handwritten digit recognition and dictionary learning for object recognition, with competitive empirical results.

## 2   Related Work

**Restricted Boltzmann Machines.**   A restricted Boltzmann machine (RBM) [8] is a bipartite Markov random field with an input layer $\mathbf{x} \in \mathbb{R}^I$ and a latent layer $\mathbf{z} \in \mathbb{R}^J$ (see Figure 1). The layers are connected by undirected weights $\mathbf{W} \in \mathbb{R}^{I \times J}$. Each unit also receives input from a bias parameter $b_j$ or $c_i$. The joint configuration of binary states $\{\mathbf{x}, \mathbf{z}\}$ has an energy given by:

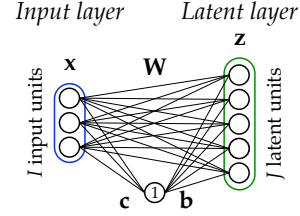

*Input layer*     *Latent layer*

Figure 1: Structure of the RBM.

$$E(\mathbf{x}, \mathbf{z}) = -\mathbf{z}^\top \mathbf{W} \mathbf{x} - \mathbf{b}^\top \mathbf{z} - \mathbf{c}^\top \mathbf{x}. \qquad (1)$$

The probability assigned to $\mathbf{x}$ is given by:

$$P(\mathbf{x}) = \frac{1}{Z} \sum_{\mathbf{z}} \exp(-E(\mathbf{x}, \mathbf{z})), \qquad Z = \sum_{\mathbf{x}} \sum_{\mathbf{z}} \exp(-E(\mathbf{x}, \mathbf{z})), \qquad (2)$$

where $Z$ is known as the partition function, which normalizes $P(\mathbf{x})$ to a valid distribution. The units in a layer are conditionally independent with distributions given by logistic functions:

$$P(\mathbf{z}|\mathbf{x}) = \prod_j P(z_j|\mathbf{x}), \qquad P(z_j|\mathbf{x}) = 1/(1 + \exp(-\mathbf{w}_j^\top \mathbf{x} - b_j)), \qquad (3)$$

$$P(\mathbf{x}|\mathbf{z}) = \prod_i P(x_i|\mathbf{z}), \qquad P(x_i|\mathbf{z}) = 1/(1 + \exp(-\mathbf{w}_i \mathbf{z} - c_i)). \qquad (4)$$

This enables the model to be sampled via alternating Gibbs sampling between the two layers. To estimate the maximum likelihood of the data distribution $P(\mathbf{x})$, the RBM is trained by taking the gradient of the log probability of the input data with respect to the parameters:

$$\frac{\partial \log P(\mathbf{x})}{\partial w_{ij}} \approx \langle x_i z_j \rangle_0 - \langle x_i z_j \rangle_N, \qquad (5)$$

where $\langle \cdot \rangle_t$ denotes the expectation under the distribution at the $t$-th sampling of the Markov chain. The first term samples the data distribution at $t = 0$, while the second term approximates the equilibrium distribution at $t = \infty$ using the contrastive divergence method [9] by using a small and finite number of sampling steps $N$ to obtain a distribution of reconstructed states at $t = N$. RBMs have also been regularized to produce sparse representations [10, 11].

**Supervised Restricted Boltzmann Machines.**   To introduce class labels to the RBM, a one-hot coded output vector $\mathbf{y} \in \mathbb{R}^C$ is defined, where $y_c = 1$ *iff* $c$ is the class index. Another set of weights $\mathbf{V} \in \mathbb{R}^{C \times J}$ connects $\mathbf{y}$ with $\mathbf{z}$. The two vectors are concatenated to form a new input vector $[\mathbf{x}, \mathbf{y}]$ for the RBM, which is linked to $\mathbf{z}$ through $[\mathbf{W}^\top, \mathbf{V}^\top]$, as shown in Figure 2. This supervised RBM models the joint distribution $P(\mathbf{x}, \mathbf{y})$. The energy function of this model can be extended to

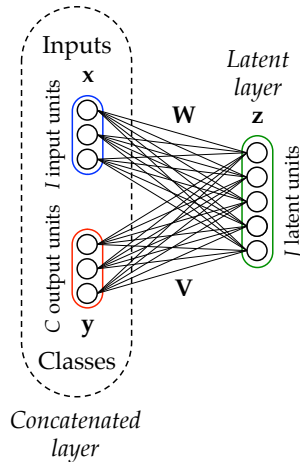

$$E(\mathbf{x}, \mathbf{y}, \mathbf{z}) = -\mathbf{z}^\top \mathbf{W} \mathbf{x} - \mathbf{z}^\top \mathbf{V} \mathbf{y} - \mathbf{b}^\top \mathbf{z} - \mathbf{c}^\top \mathbf{x} - \mathbf{d}^\top \mathbf{y} \quad (6)$$

The conditional distribution of the concatenated vector is now:

$$P(\mathbf{x}, \mathbf{y}|\mathbf{z}) = P(\mathbf{x}|\mathbf{z}) P(\mathbf{y}|\mathbf{z}) = \prod_i P(x_i|\mathbf{z}) \prod_c P(y_c|\mathbf{z}), \quad (7)$$

where $P(x_i|\mathbf{z})$ is given in Equation 4 and the outputs $y_c$ may either be logistic units or the softmax units. The RBM may again be trained using contrastive divergence algorithm [9] to approximate the maximum likelihood of joint distribution.

*Concatenated layer*

Figure 2: A supervised RBM jointly models inputs and outputs. Biases are omitted for simplicity.

During inference, only $\mathbf{x}$ is given and $\mathbf{y}$ is set at a neutral value, which makes this part of the RBM 'noisy'. The objective is to use $\mathbf{x}$ to 'denoise' $\mathbf{y}$ and obtain the prediction. This can be done by several iterations of alternating Gibbs sampling. If the number of classes is huge, the number of input units need to be huge to maintain a high signal to noise ratio. Larochelle and Bengio [12] suggested to couple this generative model $P(\mathbf{x}, \mathbf{y})$ with a discriminative model $P(\mathbf{y}|\mathbf{x})$, which can help alleviate this issue. However, if the objective is to train a deep network, then with ever new layer, the previous $\mathbf{V}$ has to be discarded and retrained. It may also not be desirable to use a discriminative criterion directly from the outputs, especially in the initial layers of the network.

**Deep Belief Networks.** Deep belief networks (DBN) [1] are probabilistic graphical models made up of a hierarchy of stochastic latent variables. Being universal approximators [13], they have been applied to a variety of problems such as image and video recognition [1, 14], dimension reduction [15]. It follows a two-phase training strategy of unsupervised greedy pre-training followed by supervised fine-tuning.

For unsupervised pre-training, a stack of RBMs is trained greedily from the bottom-up, with the latent activations of each layer used as the inputs for the next RBM. Each new layer RBM models the data distribution $P(\mathbf{x})$, such that when higher-level layers are sufficiently large, the variational bound on the likelihood always improves [1]. A popular method for supervised fine-tuning backpropagates the error given by $P(\mathbf{y}|\mathbf{x})$ to update the network's parameters. It has been shown to perform well when initialized by first learning a model of input data using unsupervised pre-training [15].

An alternative supervised method is a generative model that implements a supervised RBM (Figure 2) that models $P(\mathbf{x}, \mathbf{y})$ at the top layer. For training, the network employs the up-down back-fitting algorithm [1]. The algorithm is initialized by untying the network's recognition and generative weights. First, a stochastic bottom-up pass is performed and the generative weights are adjusted to be good at reconstructing the layer below. Next, a few iterations of alternating sampling using the respective conditional probabilities are done at the top-level supervised RBM between the concatenated vector and the latent layer. Using contrastive divergence the RBM is updated by fitting to its posterior distribution. Finally, a stochastic top-down pass adjusts bottom-up recognition weights to reconstruct the activations of the layer above.

In this work, we extend the existing DBN training strategy by having an additional supervised training phase before the discriminative error backpropagation. A top-down regularization of the network's parameters is proposed. The network is optimized globally so that the inputs gradually map to the output through the layers. We also retain the simple method of using gradient descent to update the weights of the RBMs and retain the same convention for generative RBM learning.

## 3 Top-Down RBM Regularization: The Building Block

We regularize RBM learning with targets obtained by sampling from higher-level representations.

**Generic Cross-Entropy Regularization.** The aim is to construct a top-down regularized building block for deep networks, instead of combining the optimization criteria directly [12], which is done for the supervised RBM model (Figure 2). To give control over individual elements in the latent vector, one way to manipulate the representations is to point-wise bias the activations for each latent variable $j$ [11]. Given a training dataset $\mathcal{D}_{train}$, a regularizer based on the cross-entropy loss can be defined to penalize the difference between the latent vector $\mathbf{z}$ and a target vector $\hat{\mathbf{z}}$:

$$\mathcal{L}_{RBM+reg}(\mathcal{D}_{train}) = - \sum_{k=1}^{|\mathcal{D}_{train}|} \log P(\mathbf{x}_k) - \alpha \sum_{k=1}^{|\mathcal{D}_{train}|} \sum_{j=1}^{J} \log P(\hat{z}_{jk}|z_{jk}). \tag{8}$$

The update rule of the cross-entropy-regularized RBM can be modified to:

$$\Delta w_{ij} \propto \langle x_i s_j \rangle_0 - \langle x_i z_j \rangle_N, \tag{9}$$

where

$$s_j = (1 - \lambda) z_j + \lambda \hat{z}_j \tag{10}$$

is the merger of the latent and target activations used to update the parameters. Here, the influences of $\hat{z}_j$ and $z_j$ are regulated by parameter $\lambda$. If $\lambda = 0$ or when the activationes match (i.e. $z_j = \hat{z}_j$), then the parameter update is exactly that the original contrastive divergence learning algorithm.

**Building Block.** The same principle of regularizing the latent activations can be used to combine signals from the bottom-up and top-down. This forms the building block for optimizing a DBN with top-down regularization. The basic building block is a three-layer structure consisting of three consecutive layers: the previous $\mathbf{z}_{l-1} \in \mathbb{R}^I$, current $\mathbf{z}_l \in \mathbb{R}^J$ and next $\mathbf{z}_{l+1} \in \mathbb{R}^H$ layers. The layers are connected by two sets of weight parameters $\mathbf{W}_{l-1}$ and $\mathbf{W}_l$ to the previous and next layers respectively. For the current layer $\mathbf{z}_l$, the bottom-up representations $\mathbf{z}_{l,l-1}$ are sampled from the previous layer $\mathbf{z}_{l-1}$ through weighted connections $\mathbf{W}_{l-1}$ with:

$$P(z_{l,l-1,j} \mid \mathbf{z}_{l-1}; \mathbf{W}_{l-1}) = 1/(1 + \exp(-\mathbf{w}_{l-1,j}^\top \mathbf{z}_{l-1} - b_{l,j})), \tag{11}$$

where the two terms in the subscripts of a sampled representation $\mathbf{z}_{dest,src}$ refer to the destination (*dest*) and source (*src*) layers respectively. Meanwhile, sampling from the next layer $\mathbf{z}_{l+1}$ via weights $\mathbf{W}_l$ drives the top-down representations $\mathbf{z}_{l,l+1}$:

$$P(z_{l,l+1,j} \mid \mathbf{z}_{l+1}; \mathbf{W}_l) = 1/(1 + \exp(-\mathbf{w}_{l,j}\mathbf{z}_{l+1} - c_{l,j})). \tag{12}$$

The objective is to learn the RBM parameters $\mathbf{W}_{l-1}$ that map from the previous layer $\mathbf{z}_{l-1}$ to the current latent layer $\mathbf{z}_{l,l-1}$, by maximizing the likelihood of the previous layer $P(\mathbf{z}_{l-1})$ while considering the top-down samples $\mathbf{z}_{l,l+1}$ from the next layer $\mathbf{z}_{l+1}$ as target representations. The loss function for a network with $L$ layers can be broken down as:

$$\mathcal{L}_{DBN+topdown} = \sum_{l=2}^{L} \mathcal{L}_{l,RBM+topdown} \tag{13}$$

where the cross-entropy regularization the loss function for the layer is

$$\mathcal{L}_{l,RBM+topdown} = - \sum_{k=1}^{|\mathcal{D}_{train}|} \log P(\mathbf{z}_{l-1,k}) - \alpha \sum_{k=1}^{|\mathcal{D}_{train}|} \sum_{j=1}^{J} \log P(z_{l,l+1,jk} | z_{l,l-1,jk}). \tag{14}$$

This results in the following gradient descent:

$$\Delta w_{l-1,ij} = \varepsilon \left( \langle z_{l-1,l-2,i} s_{l,j} \rangle_0 - \langle z_{l-1,l,i} z_{l,l-1,j} \rangle_N \right), \tag{15}$$

where

$$s_{l,jk} = \underbrace{(1 - \lambda_l)\, z_{l,l-1,jk}}_{\text{Bottom-up}} + \underbrace{\lambda_l\, z_{l,l+1,jk}}_{\text{Top-down}}, \tag{16}$$

is the merged representation from the bottom-up and top-down signals (see Figure 3), weighted by hyperparameter $\lambda_l$. The bias towards one source of signal can be adjusted by selecting an appropriate $\lambda_l$. Additionally, the alternating Gibbs sampling, necessary for the contrastive divergence updates, is performed from the unbiased bottom-up samples using Equation 11 and a symmetric decoder:

$$P(z_{l-1,l,j} = 1 \mid \mathbf{z}_{l,l-1}; \mathbf{W}_{l-1}) = 1/(1 + \exp(-\mathbf{w}_{l-1,i}\mathbf{z}_{l,l-1} - c_{l-1,j})). \tag{17}$$

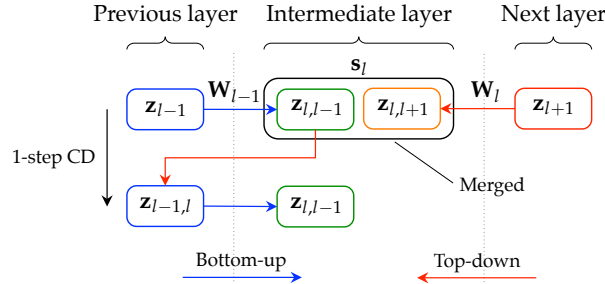

Figure 3: The basic building block learns a bottom-up latent representation regularized by top-down signals. Bottom-up $\mathbf{z}_{l,l-1}$ and top-down $\mathbf{z}_{l,l+1}$ latent activations are sampled from $\mathbf{z}_{l-1}$ and $\mathbf{z}_{l+1}$ respectively. They are merged to get the modified activations $\mathbf{s}_l$ used for parameter updates. Reconstructions independently driven from the input signals form the Gibbs sampling Markov chain.

# 4 Globally-Optimized Deep Belief Networks

**Forward-Backward Learning Strategy.** In the DBN, RBMs are stacked from the bottom-up in a greedy layer-wise manner, with each new layer modeling the posterior distribution of the previous layer. Similarly, regularized building blocks can also be used to construct the regularized DBN (Figure 4). The network, as illustrated in Figure 4(a), comprises of a total of $L - 1$ RBMs. The network can be trained with a forward and backward strategy (Figure 4(b)). It integrates top-down regularization with contrastive divergence learning, which is given by alternating Gibbs sampling between the layers (Figure 4(c)).

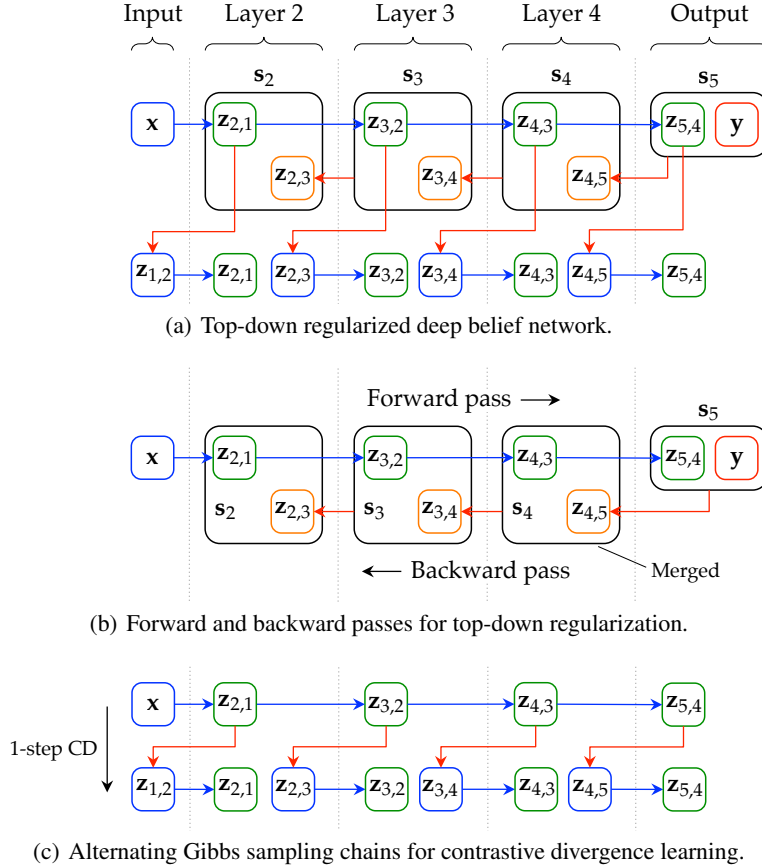

(a) Top-down regularized deep belief network.

(b) Forward and backward passes for top-down regularization.

(c) Alternating Gibbs sampling chains for contrastive divergence learning.

Figure 4: Constructing a top-down regularized deep belief network (DBN). All the restricted Boltzmann machines (RBM) that make up the network are concurrently optimized. (a) The building blocks are connected layer-wise. Both bottom-up and top-down activations are used for training the network. (b) Activations for the top-down regularization are obtained by sampling and merging the forward pass and the backward pass. (c) From the activations of the forward pass, the reconstructions can be obtained by performing alternating Gibbs sampling with the previous layer.

In the forward pass, given the input features, each layer $\mathbf{z}_l$ is sampled from the bottom-up, based on the representation of the previous layer $\mathbf{z}_{l-1}$ (Equation 11). The top-level vector $\mathbf{z}_L$ is activated with the softmax function. Upon reaching the output layer, the backward pass begins. The activations $\mathbf{z}_L$ are combined with the output labels $\mathbf{y}$ to produce $\mathbf{s}_L$ given by

$$s_{L,ck} = (1 - \lambda_L)z_{L,L-1,ck} + \lambda_L y_{ck}, \tag{18}$$

The merged activations $\mathbf{s}_l$ (Equation 16), which besides being used for parameter updates, have a second role of activating the lower layer $\mathbf{z}_{l-1}$ from the top-down:

$$P(z_{l-1,l,j} \mid \mathbf{s}_l; \mathbf{W}_l) = 1/1 + \exp(-\mathbf{w}_{l-1,j}\mathbf{s}_l - c_{l-1,j}). \tag{19}$$

This is repeated until the second layer is reached ($l = 2$) and $\mathbf{s}_2$ is computed.

Top-down sampling encourages the class-based invariance of the bottom-up representations. However, sampling from the top-down, with the output vector $\mathbf{y}$ as the only source will result in only one activation pattern per class. This is undesirable, especially for the bottom layers, which should have representations more heavily influenced by bottom-up data. By merging the top-down representations with the bottom-up ones, the representations will encode both instance-based variations and class-based variations. In the last layer, we typically set $\lambda_L$ as 1, so that the final RBM given by $\mathbf{W}_{L-1}$ learns to map to the class labels $\mathbf{y}$. Backward activation of $\mathbf{z}_{L-1,L}$ is a class-based invariant representation obtained from $\mathbf{y}$ and used to regularize $\mathbf{W}_{L-2}$. All other backward activations from this point onwards are based on the merged representation from instance- and class-based representations.

**Three-Phase Learning Procedure.** After greedy learning models $P(\mathbf{x})$ and the top-down regularized forward-backward learning is executed. The eventual goal of the network is to be able to give a prediction of $P(\mathbf{y}|\mathbf{x})$. This suggest that the network can adopt a three-phase strategy for training, whereby the parameters learned in one phase initializes the next, as follows:

- *Phase 1 – Unsupervised Greedy.* The network is constructed by greedily learning a new unsupervised RBM on top of the existing network. To enhance the representations, various regularizations, such as sparsity [10], can be applied. The stacking process is repeated for $L - 2$ RBMs, until layer $L - 1$ is added to the network.

- *Phase 2 – Supervised Regularized.* This phase begins by connecting the $L - 1$ to a final layer, which is activated by the softmax activation function for a classification problem. Using the one-hot coded output vector $\mathbf{y} \in \mathbb{R}^C$ as its target activations and setting $\lambda_L$ to 1, the RBM is learned as an associative memory with the following update:

$$\Delta w_{L-1,ic} \propto \langle z_{L-1,L-2,i}\, y_c \rangle_0 - \langle z_{L-1,L,i}\, z_{L,L-1,c} \rangle_N. \tag{20}$$

  This final RBM, together with the other RBMs learned from Phase 1, form the initialization for the top-down regularized forward-backward learning algorithm. This phase is used to fine-tune the network using generative learning, and binds the layers together by aligning all the parameters of the network with the outputs.

- *Phase 3 – Supervised Discriminative.* Finally, the supervised error backpropagation algorithm is used to improve class discrimination in the representations. Backpropagation can also be described in two passes. In the forward pass, each layer is activated from the bottom-up to obtain the class predictions. The classification error is then computed based on the groundtruth and the backward pass performs gradient descent on the parameters by backpropagating the errors through the layers from the top-down.

From Phase 1 to Phase 2, the form of the parameter update rule based on gradient descent does not change. Only that top-down signals are also taken into account. Essentially, the two phases are performing a variant of the contrastive divergence algorithm. Meanwhile, from Phase 2 to Phase 3, the inputs to the phases ($\mathbf{x}$ and $\mathbf{y}$) do not change, while the optimization function is modified from performing regularization to being completely discriminative.

## 5 Empirical Evaluation

In this work, the proposed deep learning strategy and top-down regularization method were evaluated and analyzed using the MNIST handwritten digit dataset [16] and the Caltech-101 object recognition dataset [17].

### 5.1 MNIST Handwritten Digit Recognition

The MNIST dataset contains images of handwritten digits. The task is to recognize a digit from 0 to 9 given a $28 \times 28$ pixel image. The dataset is split into $60,000$ images used for training and $10,000$ test images. Many different methods have used this dataset to perform evaluation on classification performances, specifically the DBNN [1]. The basic version of this dataset, with neither preprocessing nor enhancements, was used for the evaluation. A five-layer DBN was setup to have the same topography as evaluated in [1]. The number of units in each layer, from the first to the last layer, were 784, 500, 500, 2000 and 10, in that order. Five architectural setups were tested:

1. Stacked RBMs with up-down learning (original DBN reported in [1]),

2. Stacked RBMs with forward-backward learning and backpropagation,

3. Stacked sparse RBMs [11] with forward-backward learning and backpropagation, and

4. Stacked sparse RBMs [11] with backpropagation, and

5. Forward-backward learning from random weights.

In the phases 1 and 2, we followed the evaluation procedure of Hinton et al. [1] by initially using $44,000$ training and $10,000$ validation images to train the network before retraining it with the full training set. In phase 3, sets of $50,000$ and $10,000$ images were used as the initial training and validation sets. After model selection, the network was retrained on the training set of $60,000$ images.

To simplify the parameterization for the forward-backward learning in phase 2, the top-down modulation parameter $\lambda_l$ across the layers were controlled by a single parameter $\gamma$ using the function:

$$\lambda_l = |l-1|^\gamma/(|l-1|^\gamma - |L-l|^\gamma). \tag{21}$$

where $\gamma > 0$. The top-down influence for a layer $l$ is also dependent on its relative position in the network. The function assigns $\lambda_l$ such that the layers nearer to the input will have stronger influences from the input, while the layers near the output will be biased towards the output. This distance-based modulation of their influences enables a gradual mapping between the input and output layers.

Our best performance was obtained using setting 3, which got an error rate of $0.91\%$ on the test set. Figure 5 shows the 91 wrongly classified test examples for this setting. When initialized with the conventional RBMs but fine-tuned with forward-backward learning and error backpropagation, the score was $0.98\%$. As a comparison, the conventional DBN obtained an error rate of $1.25\%$. Directly optimizing the network from random weights produced an error of $1.61\%$, which is still fairly decent, considering that the network was optimized globally from scratch. For each setup, the intermediate results for each training phase are reported in Table 1.

Overall, the results achieved are very competitive for methods with the same complexity that rely on neither convolution nor image distortions and normalization. A variant of the DBN, which focused on learning nonlinear transformations of the feature space for nearest neighbor classification [18], had an error rate of $1.0\%$. The deep convex net [19], which utilized more complex convex-optimized modules as building blocks but did not perform fine-tuning on a global network level, got a score of $0.83\%$. At the time of writing, the best performing model on the dataset gave an error rate of $0.23\%$ and used a heavy architecture of a committee of 35 deep convolutional neural nets with elastic distortions and image normalization [20].

From Table 1, we can observe that each of the three learning phases helped to improve the overall performance of the networks. The forward-backward algorithm outperforms the up-down learning of the original DBN. Using sparse RBMs [11] and backpropagation, it was possible to further improve the recognition performances. The forward-backward learning was effective as a bridge between the other two phases, with an improvement of $0.17\%$ over the setup without phase 2. The method was even as a standalone algorithm, demonstrating its potential by learning from randomly initialized weights.

Table 1: Results on MNIST after various phases of the training process.

| Setup / Learning algorithm* | | Classification error rate | | |
|---|---|---|---|---|
| Phase 1 | Phase 2 | Phase 1 | Phase 2 | Phase 3 |
| **Deep belief network** (reported in [1]) | | | | |
| 1.   RBMs | Up-down | 2.49% | 1.25% | – |
| **Proposed top-down regularized deep belief network** | | | | |
| 2.   RBMs | Forward-backward | 2.49% | 1.14% | 0.98% |
| 3.   Sparse RBMs | Forward-backward | 2.14% | 1.06% | 0.91% |
| 4.   Sparse RBMs | – | 2.14% | – | 1.08% |
| 5.   Random weights | Forward-backward | – | 1.61% | – |

*Phase 3 runs the error backpropagation algorithm whenever employed.

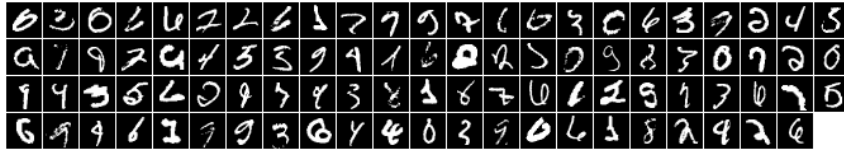

Figure 5: The 91 wrongly classified test examples from the MNIST dataset.

## 5.2 Caltech-101 Object Recognition

The Caltech-101 dataset [17] is one of the most popular datasets for object recognition evaluation. It contains $9,144$ images belonging to 101 object categories and one background class. The images were first resized while retaining their original aspect ratios, such that the longer spatial dimension was at most 300 pixels. SIFT descriptors [21] were extracted from densely sampled patches of $16 \times 16$ at 4 pixel intervals. The SIFT descriptors were $\ell_1$-normalized by constraining each descriptor vector to sum to a maximum of one, resulting in a quasi-binary feature. Additionally, SIFT descriptors from a spatial neighborhood of $2 \times 2$ were concatenated to form a macrofeature [22].

A DBN setup was used to learn a dictionary to map local macrofeatures to a mid-level representation. Two layers of RBMs were stacked to model the macrofeatures. Both RBMs were regularized with population and lifetime sparseness during training [23]. First a single RBM, which had 1024 latent variables, was trained from macrofeature. A set of $200,000$ randomly selected macrofeatures was used for training this first layer. The resulting representations of the first RBM were then concatenated within each spatial neighborhood of $2 \times 2$. The second RBM modeled this spatially aggregated representation into a higher-level representation. Another set of $200,000$ randomly selected spatially aggregated representations was used for training this RBM.

The higher-level RBM representation was associated to the image label. For each experimental trial, a set of 30 training examples per class (totaling to 3060) was randomly selected for supervised learning. The forward-backward learning algorithm was used to regularize the learning while fine-tuning the network. Finally, error backpropagation was performed to further optimize the dictionary. From these representations, max-pooling within spatial regions defined by a spatial pyramid was employed [22, 24] to obtain a single vector representing the whole image. It is also possible to employ more advanced pooling schemes [25]. A linear SVM classifier was then trained, using the same train-test split from the previous supervised learning phase.

Table 2 shows the average class-wise classification accuracy, averaged across 102 classes and 10 experimental trials. The results demonstrate a consistent improvement moving from Phase 1 to phase 3. The final accuracy obtained was 79.7%. This outperforms all existing dictionary learning methods based on a single image descriptors, with a 0.8% improvement over the previous state-of-the-art results [23, 28]. As a comparison, other existing reported dictionary learning methods that encode SIFT-based local descriptors are also included in Table 2.

Table 2: Classification accuracy on Caltech-101.

| Method / Training phase | Accuracy |
|---|---|
| Proposed top-down regularized DBN | |
| *Phase 1: Unsupervised stacking* | 72.8% |
| *Phase 2: Top-down regularization* | 78.2% |
| *Phase 3: Error backpropagation* | 79.7% |
| Sparse coding & max-pooling [22] | 73.4% |
| Extended HMAX [26] | 76.3% |
| Convolutional RBM [27] | 77.8% |
| Unsupervised & supervised RBM [23] | 78.9% |
| Gated Convolutional RBM [28] | 78.9% |

## 6 Conclusion

We proposed the notion of deep learning by gradually transitioning from being fully unsupervised to strongly discriminative. This is achieved through the introduction of an intermediate phase between the unsupervised and supervised learning phases. This notion is implemented by incorporating top-down information to DBNs through regularization. The method is easily integrated into the intermediate learning phase based on simple building blocks. It can be performed to complement greedy layer-wise unsupervised learning and discriminative optimization using error backpropagation. Empirical evaluation show that the method leads to competitive results for handwritten digit recognition and object recognition datasets.

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
