[Reviews · NeurIPS 2013]

Submitted by Assigned_Reviewer_2

This paper presents a top-down supervised regularization strategy for training deep networks. This scheme is designed to be used as an intermediate step between traditional undupervised pretraining and discriminative fine-tuning via backpropagation. The proposed optimization regularizes the DBN in a top-down fashion using a variant of contrastive divergence that backpropagates label information.

To the best of my knowledge, the idea of an intermediate supervised regularization before greedy discriminative fine-tuning is novel. The approach is sound and is formulated as a natural supervised variation of traditional unsupervised pretraining.

The approach is demonstrated with experiments on MNIST and Caltech101. The evaluation on MNIST handwritten digit recognition is the one involving the deepest empirical analysis, in terms of the number of different regularization schemes tested. It shows that the proposed intermediate step consistently improves accuracy. However, I'm personally skeptical about the statistical significance of empirical evaluations carried on such simple benchmarks, where by now clearly overtuned systems approach errors below 1%. The Caltech101 dataset is also overly simplistic for modern image recognition standards. Furthermore, the paper omits to report the comparative performance achieved by unsupervised pre-training followed by backpropagation: this is an essential baseline in order to gauge the empirical advantage of the proposed additional intermediate stage.
Summary: The paper presents an interesting, natural idea to regularize deep networks. The experiments demonstrate that there is some benefit in the approach but they are a bit inconclusive due to the overly simple datasets and the lack of comparative analysis.

Submitted by Assigned_Reviewer_5

The key idea of this paper is to use the latent vector generated from upper layers of a Deep Belief Network as target vectors, by adding a regularization cross-entropy part to the standard unsupervised training loss, so as to encourage reconstructions from bottom and top to match. Instead of the standard two-stage training of deep architectures (unsupervised layer-by-layer pretraining, then full network supervised training), training is conducted in three stages with an intermediate stage that uses this hybrid loss. Experimental results show that the intermediate stage substantially improves results on MNIST and Caltech101.
This regularization of intermediate layers by top-down generative weights shows good results; the paper is clear and shows how the proposed intuitive way of bridging unsupervised and supervised training indeed improves performance.

The paper is overall rather clear although there are some problems (see below).
Organization: the paper flows well. One minor comment:
Section 3: it would be easier for the reader to get the key idea if the intuition was given briefly at the very beginning of section 3, instead of waiting until line 192 -- in the current form, the reader is first left wondering what will be used as target vector. E.g., before the paragraph "Generic Cross-Entropy Regularization", insert a one-sentence summary saying that the main intuition is to use latent vector generated from upper layers as target vectors.
Questions/comments:
Line 044: Error backpropagation is a way to propagate the error across the lower layers of the networks; it does not say anything about being supervised or discriminative. Indeed one can very well backpropagate unsupervised reconstruction errors across layers. What makes the last training phase supervised is merely *the choice of the criterion* used for training, whose error is then backpropagated. Same goes for Line 290.
L.104: using the softmax in Eq. 8 results in the units *not* being independent, and does not correspond to the energy given in Eq.6: this is not what is depicted in Fig.2 but corresponds to a top logistic regression layer. Fig.2 and Eq.6 correspond to a logistic just as in Eq.4, with v and d instead of w and c, NOT the softmax equation given in Eq.8. Eq. 6 has exactly the same form for y and x so the resulting probability forms should not differ.
L. 262: I am a bit confused: if y is obtained by softmax, as stated l.261, is the label never used? It seems that instead the last layer z_L should be obtained by softmax (or independent binary activation, see comment above -- not sure what was used), and mixed with plugged in ground truth labels y during training. Is this what was done?
Minor comments:
writing: use more articles (e.g. "A RBM is *a* bipartite Markov random field," "This enables *the* model to be sampled")
L.44: "Given a set of trainig data, the unsupervised learning phase initializes the parameters in only one way, regardless of the ultimate classification task." This is confusing (besides, it unnecessarily narrows the context to classification as the final task). If what is meant is that the unsupervised criterion is agnostic to the final task of interest, then a straightforward formulation is better, e.g.: "The unsupervised learning phase does not take into account the ultimate task of interest, e.g. classification."
Line 052- "same gradient descent convention": gradient descent is not a convention. "procedure" better.
Line 090: missing J: V \in \R^{J \times C}.
Line 112: Bengio, not Benjio.
Line 270-278: confusing. Change presentation.
Line 344: the best score is obtained with setting 3, not 2.
Summary: This paper proposes an intuitive scheme to bridge unsupervised layer-by-layer traning and supervised whole network training stages in deep belief networks, by regularizing intermediate layer unit activations so as to skew them towards prediction made by higher layers. Experimental results show that this noticeably improves performance on MNIST and Caltech101 datasets.

Submitted by Assigned_Reviewer_6

Traditional deep belief networks (DBNs) employ a two-stage learning procedure, unsupervised layer-wise pre-training plus supervised fine tuning. This paper proposed a three-stage approach, where an intermediate step, a layer-wise training step with a two-down supervised regularization, was inserted into the procedure, making the training a smoother transition from purely unsupervised to a purely supervised stage. The idea is to introduce an additional regularization term to utilize the sampled data based on upper layers to learn the weights connecting the lower layer to the current layer. The experiments on MNIST and Caltech101 are encouraging.

The approach makes a good sense, but lacking a probabilistic explanation, therefore, is a bit ad-hoc. Furthermore, why do we need a gradual transition from the purely unsupervised training to a purely supervised training? What's the advantage? This paper should provide more insights into the important aspect, since this is the main selling point of the paper.

Overall, the paper is interesting. The idea is nice. But for a research paper, the authors should provide enough insights.
Summary: The paper presented an interesting idea. The approach is technically sensible, though a bit ad hoc. Some deeper studies to explain the main point should have been provided.
Author Feedback

Author rebuttal: We thank the three reviewers for their time in reviewing the submission and providing feedback.

In general, we are happy that the reviewers recognized our contributions to the deep learning community as being the novelty of introducing an intermediate phase that bridges the gap between the generative and discriminative training phases, as well as the implementation of this concept using a top-down regularized optimization procedure for a deep network.

Each reviewer approached the paper from a slightly different perspective and we will respond to each reviewer individually.


1) ASSIGNED_REVIEWER_2: CHOICE OF EXPERIMENTAL DATASETS

The main objective for this set of experiments was to perform a direct comparison with the original deep belief net paper using the MNIST dataset, and subsequently demonstrate the success of the method on a larger and more complex dataset.

Assigned_Reviewer_2 suggested that even the Caltech-101 dataset used is too simple. We wish to point out that although the dataset is smaller scale than other image datasets, such as the ImageNet dataset, deep learning methods not been shown to be successful on this dataset, until our attempt reported in this submission. We do not think that the Caltech-101 is as simple as claimed, mainly because of the small number of training examples (30 per class). In contrast, one of the main reasons why a larger-scale dataset, such as the ImageNet, works well with existing deep architectures is the sheer amount of training data used. In this aspect, the Caltech-101 though smaller in size, provides a far greater challenge for deep learning due to its scarceness of labeled data for training. This well-studied dataset also makes method comparison easy for a wide audience and serves as a good starting point for vision problems.

We hope that reviewer 1 considers all contributions of this paper, rather than rejecting this paper solely due to a difference in opinion of the level of challenge in the datasets.


2) ASSIGNED_REVIEWER_5: CLARITY OF POINTS

We agree with the overall assessment of Assigned_Reviewer_5. We also concur with the suggestions for an improvement in clarity for each of the highlighted point. In general, they would require minor changes in the wording and presentation of the highlighted lines. We thank the reviewer for the extra level of detail and attention when assessing this work.


3) ASSIGNED_REVIEWER_6: FURTHER INSIGHTS

The probabilistic formulation of the regularization is the comparison of the closeness of two sets of activations using the cross-entropy given by P(topdown_activation|bottomup_activation), as described in equation 15.

It is also interesting to imagine that this same optimization method can be applied to other deep networks with lesser probabilistic basis, such as deep encoder-decoder networks. For such a network, a representation balances between reconstructing the bottom layer with the matching of top-down reconstructions from the layer above.

We did not wish to speculate beyond what was given in the optimization equation and what was observed in the experiments. However, perhaps a take-home message is that the training phases may be interconnected and care should be given when training a deep network with multiple phases. In this sense, switching from a fully unsupervised phase to a strongly discriminative phase could be perceived as being ad-hoc. As such, between phases 1 and 2, we retained the generative objective, while introducing supervision, and between phases 2 and 3, we retained the labeled data. A possible insight for why this works is that phase 1 finds regions of good local minima, phase 2 refines the search space based on top-down signals, which ultimately helps the search for a good solution based on the discriminative criterion of the classification task.